# Mitochondrial Protein Homeostasis and Cardiomyopathy

**DOI:** 10.3390/ijms23063353

**Published:** 2022-03-20

**Authors:** Emily Wachoski-Dark, Tian Zhao, Aneal Khan, Timothy E. Shutt, Steven C. Greenway

**Affiliations:** 1Department of Cardiac Sciences, Cumming School of Medicine, University of Calgary, Calgary, AB T2N 4N1, Canada; emily.wachoskidark@ucalgary.ca; 2Libin Cardiovascular Institute, Cumming School of Medicine, University of Calgary, Calgary, AB T2N 4N1, Canada; 3Department of Biochemistry and Molecular Biology, Cumming School of Medicine, University of Calgary, Calgary, AB T2N 4N1, Canada; tian.zhao@ucalgary.ca; 4Department of Pediatrics, Cumming School of Medicine, University of Calgary, Calgary, AB T2N 4N1, Canada; khaa@ucalgary.ca; 5Alberta Children’s Hospital Research Institute, University of Calgary, Calgary, AB T2N 4N1, Canada; 6M.A.G.I.C. Inc., Calgary, AB T2E 7Z4, Canada; 7Department of Medical Genetics, Cumming School of Medicine, University of Calgary, Calgary, AB T2N 4N1, Canada; 8Hotchkiss Brain Institute, Cumming School of Medicine, University of Calgary, Calgary, AB T2N 4N1, Canada

**Keywords:** protein homeostasis, cardiomyopathy, protein import, mitochondria, unfolded protein response, integrated stress response

## Abstract

Human mitochondrial disorders impact tissues with high energetic demands and can be associated with cardiac muscle disease (cardiomyopathy) and early mortality. However, the mechanistic link between mitochondrial disease and the development of cardiomyopathy is frequently unclear. In addition, there is often marked phenotypic heterogeneity between patients, even between those with the same genetic variant, which is also not well understood. Several of the mitochondrial cardiomyopathies are related to defects in the maintenance of mitochondrial protein homeostasis, or proteostasis. This essential process involves the importing, sorting, folding and degradation of preproteins into fully functional mature structures inside mitochondria. Disrupted mitochondrial proteostasis interferes with mitochondrial energetics and ATP production, which can directly impact cardiac function. An inability to maintain proteostasis can result in mitochondrial dysfunction and subsequent mitophagy or even apoptosis. We review the known mitochondrial diseases that have been associated with cardiomyopathy and which arise from mutations in genes that are important for mitochondrial proteostasis. Genes discussed include DnaJ heat shock protein family member C19 (*DNAJC19*), mitochondrial import inner membrane translocase subunit TIM16 (*MAGMAS*), translocase of the inner mitochondrial membrane 50 (*TIMM50*), mitochondrial intermediate peptidase (*MIPEP*), X-prolyl-aminopeptidase 3 (*XPNPEP3*), HtraA serine peptidase 2 (*HTRA2*), caseinolytic mitochondrial peptidase chaperone subunit B (*CLPB*) and heat shock 60-kD protein 1 (HSPD1). The identification and description of disorders with a shared mechanism of disease may provide further insights into the disease process and assist with the identification of potential therapeutics.

## 1. Introduction

Cardiac muscle relies heavily on mitochondrial oxidative phosphorylation (OXPHOS) to meet the constant energy demands that are needed for myofilament contraction and electrical function [1]. There are many examples demonstrating that abnormal mitochondrial function caused by genetic disorders can lead to diseases of the heart muscle. 

### 1.1. Cardiomyopathy in Humans

The cardiomyopathies are a diverse group of disorders that can be defined by their etiology (e.g., genetic or acquired) or by the morphology of the heart (e.g., dilated or hypertrophic) [2,3,4,5]. The most common forms are classified as hypertrophic cardiomyopathy (HCM), dilated cardiomyopathy (DCM), restrictive cardiomyopathy (RCM), arrhythmogenic cardiomyopathy (ACM) or left ventricular noncompaction (LVNC) [4]. All of these cardiomyopathies can have varying severity and variable age of onset. In HCM, the myocardium thickens and stiffens, increasing intracardiac pressures and frequently causes outflow tract obstruction leading to impaired cardiac output [5]. DCM is the phenotypic opposite of HCM, with the ventricle(s) becoming thin-walled and dilated with poor contractility [6]. Notably, it is well recognized that patients with HCM can progress to a dilated phase that closely resembles idiopathic DCM [7,8]. LVNC is a heterogeneous disorder characterized by a hypertrabeculated myocardium that is variably associated with dysfunction [4]. 

### 1.2. Causes of Cardiomyopathy

Many familial sarcomere protein mutations and neuromuscular disorders associated with mitochondrial structural dysfunction, such as Friedreich ataxia and Duchenne Muscular Dystrophy, are accompanied by cardiomyopathy [9,10]. Specific defects related to substrate utilization through OXPHOS, mitochondrial DNA deletion syndromes, fatty acid oxidation defects and a number of other inborn errors of metabolism can also present with cardiomyopathy [11,12]. The relationship between acute stress (e.g., myocardial infarction and ischemia-reperfusion injury) and abnormal energetics and cardiac dysfunction is well studied. However, the role of cardiomyopathy related to chronic dysfunction of mitochondrial protein homeostasis, or proteostasis, remains relatively understudied [13,14]. 

Considering the immense energetic demands of the heart, dysfunction in the electron transport chain (ETC), an essential series of protein complexes for successful OXPHOS, is often linked to heart disease [15,16]. The ETC is comprised of four complexes that couple redox reactions to create an electrochemical gradient that can then be used to generate ATP. Deficiency of these complexes, or mutations in the genes encoding their subunits, are often associated with cardiomyopathy. Complex I deficiency is primarily associated with HCM, while the deficiency of complexes II, III, or IV are variably associated with HCM or DCM [16]. 

Proper function of the ETC is partially dependent on the availability and distribution of metal ions within mitochondria [17]. Iron metalloproteins are particularly important as Fe/S clusters are found within complexes I, II, and III and play a crucial role in electron transfer [18]. Another key player is cytochrome *c* oxidase, a copper metalloprotein that serves as the final electron acceptor in the ETC [19]. Without these metalloproteins the ETC cannot properly function and meet the energetic demands of the heart.

Deficiency and dysfunction of ETC complexes not only limits ATP production but also produces excessive harmful reactive oxygen species (ROS). While ROS is regularly produced and managed under physiological conditions, an abnormal increase in ROS is associated with heart disease [20,21]. This increase in ROS can lead to oxidative stress, an inability to properly manage excess ROS, and cardiac dysfunction [21].

Imbalances in ROS production negatively affect calcium handling, which is imperative for cardiac contractility, and can impair vascular signaling which can lead to vascular dysfunction [22]. Additionally, certain protein complexes, such as the mitochondrial permeability transition pore (mPTP), are involved in calcium handling which is essential for maintaining cardiac function [23,24]. The uptake of calcium into mitochondria through the uniporter protein complex is required for OXPHOS. Thus, if calcium cannot be released from mitochondria through defects in ion channels or the mPTP caused by impaired protein import or protein handling, then defects in cardiac contractility could arise. Unsurprisingly, abnormal calcium handling is associated with both HCM and DCM [25,26,27,28].

### 1.3. Mitochondrial Encoded Proteins

Although mitochondria contain their own genome, the vast majority (>99%) of mitochondrial proteins are encoded by nuclear DNA and require import into mitochondria [29]. This process is highly influenced by the unique structure of mitochondria, comprised of an outer mitochondrial membrane (OMM) and an inner mitochondrial membrane (IMM) separated by an intermembrane space (IMS) [30]. Ions and small molecules are able to freely pass across the OMM but the remaining larger molecules, particularly proteins, must use specific translocases. The IMM is even less permeable than the OMM and requires the use of translocases for most molecules, including key proteins required for OXPHOS and energy production [31]. Of the mitochondrial proteins, only 13 are encoded by mitochondrial DNA, requiring that the rest be imported and folded to become fully functional [16]. In fact, complexes I, III, IV and ATP synthase are comprised of both nuclear- and mitochondrial- encoded proteins, while complex II is comprised solely of nuclear-encoded proteins [32]. Defects in mitochondrial protein import result in ETC deficiencies and subsequent cardiac dysfunction when the necessary proteins cannot be properly assembled into functional complexes. 

### 1.4. Consequences of Disrupted Mitochondrial Proteostasis 

Maintenance of the mitochondrial proteome is critically important to ensure proper mitochondrial function. Mitochondria maintain proteostasis through a dynamic process that regulates protein import, folding, targeting, processing, and degradation of proteins [33]. All of these critical processes must be properly coordinated to ensure mitochondrial viability.

An inability to import proteins into mitochondria can result in impaired proteostasis, which can lead to mitochondrial death if not rapidly restored [34]. Disruption of mitochondrial proteostasis places ATP production at risk and can lead to the accumulation of harmful ROS which stimulate mitochondrial fission and cellular apoptosis [34,35,36]. Cells have evolved multiple ways to address mitochondrial dysfunction and maintain mitochondrial energetics, depending on the source and severity of damage [37].

Key cellular responses include activation of the mitochondrial unfolded protein response (UPRmt), which is triggered by the presence of proteinaceous aggregates caused by protein misfolding, and the integrated stress response (ISR), a regulatory process which responds to numerous pathological insults and assists the cell in restoring proteostasis [37]. These and other stress responses are activated in similar ways but use different downstream processes to address mitochondrial dysfunction. For example, transcription factor C/EBP homologous protein (CHOP), activating transcription factor 5 (ATF5) and activating transcription factor 4 (ATF4) are all essential for activation of the ISR but are also imperative to UPRmt activation [38,39]. However, the ISR can activate expression of multiple genes based on the intensity and type of stressors, while the UPRmt primarily activates genes encoding proteins that address protein misfolding and degradation [40]. Nonetheless, when damage cannot be attenuated through these pathways, cardiomyopathy can develop [1].

This review describes a subset of known disorders of mitochondrial proteostasis that are associated with cardiomyopathy. Genes known to be associated with human disease and involved in mitochondrial protein import were identified and then screened for reports of associated human cardiomyopathy. Shared pathways and pathologies and new areas for research are identified. In the next sections, we will review the critical processes required to maintain mitochondrial proteostasis including protein import, folding and degradation and introduce some of the key players mediating these processes.

## 2. Mitochondrial Protein Import

The import of proteins into mitochondria relies on membrane translocase complexes to transport and sort proteins into their appropriate compartment. Proteins destined for either the OMM, IMS, IMM or the mitochondrial matrix contain specific targeting sequences to ensure their appropriate localization [30]. There are multiple pathways that proteins can follow depending on their intended destination [41]. The presequence pathway is used for proteins containing unique signals, indicating that they are destined for the IMM or the matrix. The carrier pathway will insert select hydrophobic proteins into the IMM, whereas the oxidative folding pathway is used for proteins found in the IMS. Lastly, the internal targeting signal pathway is used for β-barrel proteins or α-helical proteins that reside in or on the OMM [30,42].

### 2.1. Presequence Pathway

The presequence pathway includes two translocase complexes. The translocase of the outer membrane (TOM) and the translocase of the inner membrane (TIM) import and sort proteins with an α-helical region at the N-terminus [41]. The TOM complex is comprised of three major subunits. Tom20 and Tom22 bind to hydrophobic and positively charged residues, respectively, while Tom40 forms the central β-barrel channel through which proteins move [30,43]. Once through the TOM complex, a subset of hydrophobic membrane proteins destined for the IMM, such as translocase subunits or carrier proteins for the transport of polar molecules and ions, are transferred to the TIM22 complex. The TIM23 complex mediates the import of all other proteins destined for the matrix [41,44]. TIM23 includes the mitochondrial heat shock protein 70 (mtHsp70) which uses an ATP-dependent process to move proteins into the mitochondrial matrix. The ATPase activity of mtHsp70 is regulated by the presequence associated motor (PAM) complex, comprised of mitochondrial import inner membrane translocase subunit TIM16 (MAGMAS) and DnaJ heat shock protein family member C19 or C15 (DNAJC19 or DNAJC15), which aid in complex assembly and localization [29,35,45]. It is worth noting that DNAJC15 and DNAJC19 both positively regulate mtHsp70 activity but form distinct complexes, with DNAJC19 being essential for protein translocation while DNAJC15 serves a dispensable function [35]. Once through TIM23, the presequences of the preproteins destined for the IMM or matrix are removed by the mitochondrial processing peptidase (MPP) (Figure 1) [41]. Following cleavage by MPP some proteins require secondary cleavage for added stability, which is accomplished by the mitochondrial intermediate peptidase (MIP), encoded by *MIPEP*, or X-prolyl-aminopeptidase 3 (XPNPEP3), encoded by *XPNPEP3* [46]. MIP removes an additional N-terminal octapeptide while XPNPEP3 removes a single destabilizing N-terminal amino acid such as tyrosine or phenylalanine [46,47].

### 2.2. Internal Targeting Signal Pathway

Preproteins that follow the internal targeting signal pathway usually have multiple hydrophobic sequences within the protein structure that resemble the N-terminal targeting sequences but are not cleaved [41]. For proteins destined for the OMM, the sorting and assembly machinery (SAM) complex will integrate those proteins with β-barrels into the OMM [31]. Notably, this same machinery is required for helping the TOM subunits assemble. Thus, the SAM complex also plays an important role in ensuring the proper function of downstream protein import [43]. 

## 3. Mitochondrial Chaperones and Proteases 

Chaperones assist in the process of folding imported proteins into their mature and fully functional forms. Co-chaperones contain domains that enable specific protein–protein interactions in order to assist chaperones in their function [48]. Some co-chaperones are known as chaperone mediators as they act to modulate the activity of chaperones. Chaperones are necessary for cell viability and their loss can result in the disruption of mitochondrial proteostasis due to the accumulation of misfolded and degraded proteins [45,48,49,50].

Proteases play an important role in the processing of incoming mitochondrial proteins and the selective removal of misfolded proteins [51,52]. In some cases, protein degradation serves as a mediator for certain signaling cascades, some of which lead to mitophagy or apoptosis [52]. Disturbances in mitochondrial proteases can result in the accumulation of misfolded proteins, resulting in perturbed proteostasis [51].

Chaperones and proteases also play a role in mitochondrial stress responses [53,54]. During activation of the UPRmt, we see an upregulation of genes encoding chaperone proteins (in order to assist with the refolding of proteins) and an upregulation of proteases (which can degrade protein aggregates) which enable mitochondria to regain homeostasis [53]. The best characterized chaperones include mitochondrial heat shock protein 70 (mtHsp70), heat shock protein 60 (Hsp60), and heat shock protein 10 (Hsp10) [55]. Caseinolytic mitochondrial peptidase chaperone subunit B (CLPB) serves to solubilize aggregated proteins, while lon peptidase 1 (LONP1) and caseinolytic mitochondrial matrix peptidase proteolytic subunit (ClPXP) act to degrade misfolded proteins [55].

## 4. Consequences of Disrupted Mitochondrial Proteostasis

Perturbed proteostasis through defective import or the assembly of protein complexes, such as respiratory chain super complexes, can lead to disruptions in OXPHOS and cause oxidative stress [21]. Oxidative stress not only limits mitochondrial capacity to produce ATP but also produces excessive and toxic ROS which can further exacerbate mitochondrial dysfunction [39,50]. Although the function of the electron transport chain is not perfect and is associated with the production of small amounts of ROS under physiological conditions, disruptions in proteostasis lead to increased ROS production with consequences for mitochondrial and cellular function [21].

## 5. Diagnosing Mitochondrial Disease

Mitochondrial dysfunction is apparent in numerous diseases beyond cardiovascular disorders [56]. For example, Warburg’s theory postulates that cancer is a mitochondrial metabolic disease due to insufficient mitochondrial respiration [57]. Additionally, type 2 diabetes has been associated with numerous mutations in mitochondrial DNA causing mitochondrial dysfunction in patients [58]. Thus, it is evident that mitochondrial diseases are a heterogenous group of disorders. Variability in clinical phenotype and age of onset often creates difficulty when trying to diagnose these diseases [56]. While many disorders will show overlap in clinical characteristics, for the mitochondrial diseases there tends to be phenotypic variation within patient populations, making timely diagnosis difficult [56,59].

## 6. Genes Linked to Mitochondrial Protein Import and Cardiomyopathy

A subset of disorders is known to arise from mutations in genes that play a role in protein import into the mitochondrial matrix. Starting first with genes involved in mitochondrial protein import, we will look at *DNAJC19, MAGMAS* and *TIMM50* (Table 1). These three genes directly encode subunits associated with the TIM23 translocase and are thus directly associated with mitochondrial protein import into the IMM (Figure 1) [45,60,61].

### 6.1. DNAJC19 & DCMA

*DNAJC19* (DNAJ/HSP40 homolog, subfamily C, member 19) encodes DNAJC19, a DNAJ-family protein that localizes to the IMM [45]. While the exact role of DNAJC19 is unknown, based on its homology to the yeast PAM18 protein that interacts with PAM16 to mediate mitochondrial protein import via the TIM23 complex, it was initially thought that mammalian DNAJC19 may interact with MAGMAS in a similar way (Figure 1) [41,45]. Furthermore, an in vitro study found that both DNAJC19 and MAGMAS play critical roles in protein translocation through stimulation and inhibition of mtHsp70 ATPase activity [35]. Therefore, deficiency of DNAJC19 could result in a detrimental loss of mitochondrial proteostasis through perturbed protein import. However, there have been conflicting findings regarding the role of DNAJC19 with one group reporting that DNAJC19 is involved in remodeling of the mitochondrial membrane phospholipid cardiolipin although this has not been a universal finding in patient cells [62,63,64].

Loss of DNAJC19 results in the dilated cardiomyopathy with ataxia syndrome (DCMA) [45]. This rare mitochondrial disorder was first identified in the Canadian Dariusleut Hutterite population of southern Alberta, though isolated cases have since been described worldwide (Figure 2) [45,65,66,67,68]. All pathogenic DCMA variants described to date lead to the premature truncation of the protein DNAJC19 [45,65,66,67,68]. DCMA is clinically characterized by increased levels of 3-methylglutaconic acid (3-MGC) in blood and urine, DCM, conduction defects, hypotonia, cerebellar ataxia and developmental delay [45,65,69]. Some patients also exhibit growth retardation, neurodevelopmental issues and cataracts. Other notable features reported include optic atrophy and genital abnormalities in males [45,65,66,67,68,69]. In the Hutterite population in southern Alberta, homozygosity for a founder mutation (c.130-1G > C, IVS3-1G > C) in the *DNAJC19* gene has been shown to be causative for the disorder [45,69]. However, despite this genetic similarity, the phenotype of DCMA varies widely between patients and even within families [70]. DCMA has some phenotypic overlap with Barth syndrome, a disease caused by mutations in phospholipid-lysophospholipid transacylase, tafazzin (*TAZ*), encoding tafazzin, which is involved in the processing of cardiolipin [62,71,72]. Despite not knowing the exact function(s) of DNAJC19, it is clear that pathogenic variants in *DNAJC19* impair mitochondrial function and could result in perturbed proteostasis. However, since DCMA is relatively understudied, this remains to be experimentally confirmed [63,64,66,67,68].

### 6.2. MAGMAS & SMDMDM

One of the subunits of TIM23 is MAGMAS (mitochondrial import inner membrane translocase subunit TIM16), encoded by *MAGMAS*, which serves an important role in protein import [60,73]. MAGMAS was found to inhibit ATPase activity of mtHsp70, while DNAJC19 likely plays an activating role [60]. Both MAGMAS and DNAJC19 have been shown to be essential for the formation and protein import activity of TIM23 [35,60]. Unsurprisingly, a loss of protein import caused by impaired TIM23 function results in activation of the UPRmt, suggesting disrupted mitochondrial proteostasis [74].

It is important to note that while mutations in *DNAJC19* are associated with DCM, mutations in *MAGMAS* are not always associated with cardiomyopathy [75,76,77]. The Megarbane-Dagher-Melki type of spondylometaphyseal dysplasia (SMDMDM) results from mutations in *MAGMAS*. This disorder was initially described in two siblings of a Lebanese family who presented with severe platyspondyly (widening of the vertebral bodies), short stature and developmental delay but who did not have cardiomyopathy [78]. Later, a second, and reportedly unrelated, Lebanese family with similar clinical findings was characterized. In contrast to the first family, this pair of siblings had severe DCM, leading to heart failure by the age of 2 years [75]. The same missense variant in *MAGMAS* (N76D) was discovered in both families, even though the cardiac phenotype was discordant [77]. More recently, an individual of European descent was reported with a milder phenotype of SMDMDM, resulting from a nearby variant in *MAGMAS* (Q74P). This person had mild dysmorphic features and conductive hearing loss but no cardiomyopathy [76]. 

### 6.3. TIMM50 & MGCA9 

Translocase of the inner mitochondrial membrane 50 (TIMM50), encoded by *TIMM50*, is a subunit of the TIM23 import complex and is believed to be essential for the proper import of presequence proteins into the mitochondrial matrix (Figure 1) [61]. Knockdown of TIMM50 is associated with lower levels of components of the TIM23 complex, including TIMM17A, TIMM17B, TIMM23 and DNAJC19 [61]. Interestingly, another study showed that TIMM50 deficiency resulted in the appearance of fragmented mitochondria, similar to what has been seen in cells from patients with DCMA caused by mutations in *DNAJC19* [63,64,79]. While levels of respiratory chain subunits in patient fibroblasts were not disrupted, the assembly of respiratory chain complexes was reduced along with a reduction in the oxygen consumption rate [79]. These findings suggest that mutations in TIMM50 are associated with altered mitochondrial structure and function. 

Mutations in TIMM50 lead to the clinical phenotype of 3-methylglutaconic aciduria type IX (MGCA9), an autosomal recessive disorder characterized by developmental delay, seizures, hypotonia and increased levels of 3-MGC, similar to DCMA and other 3-methylglutaconic acidurias, including type VII (MGCA7) and VIII (MGCA8), which will be discussed later [61,80,81]. One patient with MGCA9 was reported to have mild left ventricular hypertrophy (such as can be seen in HCM) [81], while two patients were reported as having DCM [79,80] with one of these patients succumbing to cardiorespiratory arrest [61]. A zebrafish model of TIMM50 depletion showed cardiac dilation [82], while a mouse knockout model of *TIMM50* showed cardiac hypertrophy [83]. Variation in cardiomyopathy could be related to the severity of oxidative stress and ROS production created by mitochondrial dysfunction [84,85].

## 7. Genes Linked to Mitochondrial Protein Processing and Cardiomyopathy

Genes involved in mitochondrial protein processing or localization within the mitochondrial matrix include *MIPEP*, *XPNPEP3* and *HTRA2* (Table 2). These genes encode peptidases and proteases that play important roles in protein processing and the maturation of mitochondrial proteins, helping to maintain mitochondrial proteostasis.

### 7.1. MIPEP & COXPD31

*MIPEP* encodes mitochondrial intermediate peptidase (MIP), which is responsible for the secondary cleavage of select proteins entering mitochondria (Figure 1). Increased expression of MIP is found in tissues with high rates of oxygen consumption such as the heart and skeletal muscle [86] and MIP in yeast is known to be important for the processing of OXPHOS complexes [87]. However, a recent study has suggested that MIP is also responsible for the processing and activation of ClpP, a protease that plays a role in mitochondrial homeostasis by degrading aggregated proteins during activation of the UPRmt [88]. This suggests a role for MIP in the maintenance of mitochondrial proteostasis by activating important proteases during mitochondrial stress.

Pathogenic variants in *MIPEP* have been described in four unrelated children with combined oxidative phosphorylation deficiency 31 (COXPD31) [89]. COXPD31 is characterized by global developmental delay, severe hypotonia and DCM [89]. Other notable patient features include facial dysmorphisms, short stature, neurodevelopmental issues (e.g., epilepsy, microcephaly) and cataracts. Additionally, there have been reports of individuals with LVNC and HCM [89]. 

### 7.2. XPNPEP3 & NPHPL1

X-prolyl aminopeptidase 3 (XPNPEP3), encoded by *XPNPEP3*, is a mitochondrial protease that is responsible for the secondary cleavage of proteins localized to the matrix (Figure 1) [90]. Specifically, XPNPEP3 removes the N-terminal amino acid from polypeptides with a penultimate proline residue [47]. It is estimated that around 25% of all proteins cleaved by MPP will undergo secondary cleavage by either XPNPEP3 or MIP [91]. Additionally, both MIP and XPNPEP3 have been implicated in ClpP activation [47,88]. Thus, secondary cleavage events by mitochondrial proteases such as XPNPEP3 and MIP are important for the maintenance of mitochondrial energetics and proteome stability. 

Nephronophthisis (NPHP) is a disease of the kidney leading to renal failure during childhood. Genes linked to NPHP encode components of cilia [92]. Pathogenic variants in *XPNPEP3*, encoding a mitochondrial protease, have been identified in NPHPL1 (nephronophthisis-like neuropathy 1), which closely resembles NPHP [93,94,95]. While XPNPEP3 is believed to be implicated in the processing of cilia proteins, it does not localize to cilia, as do proteins involved in NPHP [94]. In addition to NPHP, patients with *XPNPEP3* variants have extrarenal manifestations that could be caused by mitochondrial dysfunction rather than ciliary dysfunction [94]. Neurological features include essential tremor, hearing loss, muscle fatigue, seizures, and developmental delay. In addition, HCM and DCM were reported alongside a complex I deficiency found in two patients with NPHPL1 [94]. Importantly, while similar to NPHPL1, NPHP has not been found to present with cardiomyopathy [92]. This would suggest that the mitochondrial role of XPNPEP3 implicates the cardiac phenotype rather than its ciliary role. 

### 7.3. HTRA2 & MGCA8 

HtraA serine peptidase 2 (*HTRA2*) is a gene encoding the mitochondrial protease HTRA2 which is targeted to the IMS, where it regulates apoptosis through the induction of caspase activity [96]. Additionally, HTRA2 has been reported to serve a multifunctional role as both a chaperone and protease depending on cellular conditions [97]. In *E. coli*, the HTRA2 homolog acts as a chaperone at low temperatures and a protease at high temperatures [98]. In humans, HTRA2 was initially described as a protease but recent studies have shown an additional role for HTRA2 as a chaperone [96]. Particularly, HTRA2 was thought to assist with protein folding and preventing the aggregation of amyloidogenic peptides. This was supported by an in vitro study showing that chaperone activity of HTRA2 was not dependent of its proteolytic domain [99]. While there is still more to learn about the variable role of HTRA2, it is thought to be important during mitochondrial dysfunction as it was found to be upregulated under stress conditions such as those that activate the UPRmt [98].

Disorders of HTRA2 result in 3-MGA type VIII (MGCA8) and are associated with mitochondrial dysfunction, elevated 3-MGC, hypotonia and seizures [100]. Despite being most commonly associated with neurodegeneration, a patient with MGCA8 presented with heart failure and poor cardiac function [100]. However, mice with a cardiac-specific overexpression of HTRA2 have DCM [101], while elevated HTRA2 was reported in the left ventricle of dogs with chronic heart failure [102]. Thus, an imbalance of HTRA2 levels appears to be associated with cardiomyopathy.

## 8. Genes Linked to Mitochondrial Chaperone Activity and Cardiomyopathy

Next, we will look at genes encoding mitochondrial chaperones. These genes include *CLPB* and *HSP60* (Table 3). Mutations in these genes lead to impaired mitochondrial energetics, much like the disorders discussed in the previous sections, suggesting that there are multiple potential pathways through which a loss of proteostasis could lead to cardiomyopathy.

### 8.1. CLPB & MEGANN

Caseinolytic mitochondrial peptidase chaperone subunit B (CLPB), encoded by *CLPB*, is a protein disaggregase belonging to the Clp group of the AAA+ protease family. CLPB is known to localize to the IMS where it re-solubilizes aggregated proteins [103,104]. Notably, CLPB is activated by proteolytic cleavage mediated by the mitochondrial presenilins-associated rhomboid-like protein (PARL). PARL plays a role in mitochondrial quality control through regulation of apoptosis and mitochondrial dynamics such as fission and fusion [105]. Based on the role of PARL, CLPB could be subsequently activated to assist in maintaining proteostasis and mitochondrial quality control. Similarly, in *E. coli* the CLPB homolog was found to interact with the Hsp70 chaperone system, recruited by DnaJ, to aid in protein disaggregation [106]. Lastly, loss of CLPB leads to the accumulation of several TIM complex proteins involved in chaperone-mediated protein transport and IMM insertion [103]. Thus, CLPB is a mitochondrial chaperone that is linked functionally to mitochondrial protein import and proteostasis through its multifaceted role. 

MEGCANN (or 3-methylglutaconic aciduria type VII; MGCA7) is an inborn error of metabolism caused by homozygous or compound heterozygous mutations in *CLPB* [72,104,107,108,109,110,111,112,113]. MEGCANN primarily presents with elevated levels of 3-MGC, although neutropenia, neurologic deterioration (including progressive movement disorder), and cataracts are common in affected individuals [72,107,108,109]. In addition, impaired growth and skeletal features have also been reported [107,108,109]. Less common features include mild DCM and hypothyroidism [107]. Notably, impaired OXPHOS and alterations in mitochondrial morphology are not evident in patients with MEGCANN [107,109]. This would suggest that impaired proteostasis does not always correlate with impaired OXPHOS. Interestingly, impaired disaggregase activity correlates with disease severity, implying that the loss of this activity underlies the disease and could explain the wide variation in severity of patient phenotypes observed [103]. This fact could potentially explain why cardiomyopathy has only been reported in a few patients, as the severity of disease may not be sufficient to alter mitochondrial function to the point of causing cardiomyopathy. Indeed, all reports of cardiac involvement have been found in patients with a severe phenotype [107].

### 8.2. HSP60 & Hypomyelinating Leukodystrophy 4 and Spastic Paraplegia 13

Hsp60 is a chaperone encoded by the heat shock 60-kD protein 1 (*HSPD1*) gene and localizes primarily to mitochondria but can be found in the cytosol [114]. As a chaperone, the main role of Hsp60 is to assist with the folding of proteins in the mitochondrial matrix, although it also plays a role in the restoration of misfolded or denatured proteins [115]. Importantly, it was found that, without Hsp60, proteins reliant on this chaperone for folding and maturation were readily degraded and indeed triggered the UPRmt, suggesting a role for Hsp60 in mitochondrial proteostasis [54]. 

To date, disorders resulting from mutations in the *HSPD1* gene have not been associated with cardiomyopathy in humans [116]. Instead, mutations in *HSPD1* are associated with hypomyelinating leukodystrophy as well as spastic paraplegia [116,117,118,119]. Nonetheless, a recent in vivo study using mice noted that the deletion of Hsp60 resulted in a loss of proteostasis and was associated with DCM and increased apoptosis leading to death [54]. Similarly, an in vivo study using rats noted that Hsp60 played a neuroprotective role in the rostral ventrolateral medulla which regulates cardiovascular function and can lead to cardiovascular fatality during brain stem death [120]. This same study found that Hsp60 redistributed from the mitochondria to the cytosol where it exerted an anti-apoptotic effect through reduction of the cytochrome c-caspase-3 cascade [120]. Taken together, these studies in murine models show that Hsp60 does indeed have the potential to play a role in the maintenance of cardiovascular function, despite not being associated with cardiomyopathy in humans to date. While this gene is not linked to cardiomyopathy in humans, it perhaps serves as a reminder that diseases in animal models do not always replicate what is seen in humans [121,122,123,124,125]. Alternatively, it should be noted that patients with hypomyelinating leukodystrophy generally die within the first twenty years of life with many dying before the age of two years [116]. With such early mortality, it is possible that they do not survive long enough to develop a detectable phenotype of cardiomyopathy.

## 9. Cellular Responses to Abnormal Mitochondrial Proteostasis 

The majority of disorders discussed within this review are caused by mutations impacting genes that are ubiquitously expressed (e.g., *DNAJC19* and *XPNPEP3*) [45,126] or that are highly expressed in the heart, skeletal muscle and the brain (e.g., *MAGMAS*, *MIPEP*, *CLPB*, and *TIMM50*) [82,86,87,127,128]. The expression pattern of these genes points to the importance of their encoded proteins in energy production. Of all the disorders discussed, those that seem to most drastically affect proteins related to OXPHOS, such as DCMA, MGCA9 and COXPD31, frequently present with cardiomyopathy. OXPHOS dysfunction can lead to oxidative stress and subsequent increases in ROS production, which is commonly associated with cardiomyopathy [129]. Maintenance of mitochondrial proteostasis is particularly important for cellular energy production as the import, processing, and folding of incoming proteins is required for the proper formation of respiratory chain complexes [74]. A disruption of mitochondrial proteostasis can cause significant disruptions in ATP production and lead to oxidative stress and the accumulation of toxic ROS [130]. Injury to mitochondria through oxidative stress also leads to alterations in mitochondrial fission and fusion dynamics, leading to excessive mitophagy which plays a role in cardiomyopathy and heart failure [129,130,131]. Mouse models that abrogate mitochondrial fission and fusion show that this ultimately leads to complete failure of heart function [132]. This would suggest that chronically perturbed proteostasis is linked to cardiomyopathy through abnormal mitochondrial energetics and imbalances in ROS, leading to mitochondrial failure. Mitochondrial failure would subsequently result in heart failure [133,134]. 

To counterbalance this process, the cell has multiple stress responses such as the ISR and UPRmt, which serve to address a loss of mitochondrial proteostasis and limit mitochondrial stress by solubilizing aggregated proteins, refolding, and degrading proteins [53]. Activation of the ISR has shown to be effective in reducing ischemia-reperfusion (IR) injury, a condition that leads to protein misfolding and oxidative stress, similar to what is observed with perturbed protein import and preprotein processing [39,135]. The way in which the ISR addresses IR injury is through inhibiting translation of select proteins and preventing the assembly of mitochondrial protein complexes [136]. Reduction in the formation of mitochondrial respiratory chain complexes slows electron transfer and reduces ROS production, preventing further mitochondrial dysfunction [136]. Additionally, one study found that activation of the UPRmt through the upregulation of ATF5 led to cardioprotection in a mouse model of cardiac IR injury [137]. Thus, if the UPRmt serves a cardioprotective role in acute IR injury then it could also potentially play a role in cardioprotection in chronic disorders of proteostasis leading to cardiomyopathy. However, IR injury is representative of acute stress and not the chronic stress that would be observed in disorders of proteostasis. In fact, it is likely that initial activation of these stress responses in chronic conditions have the same goals as when they are activated during acute stress. Where these stress responses then differ would be in long-term activation. 

While the UPRmt is an initial adaptive stress response aimed at refolding and degrading misfolded proteins, when stress cannot be attenuated, prolonged activation of the UPRmt becomes maladaptive and can lead to excessive mitophagy and in some cases, apoptosis [138,139]. This potentially suggests that differences in the severity of cardiac involvement in certain patients may be related to the degree of activation of the UPRmt. In the disorders that frequently present with cardiomyopathy and have high mortality rates, cells may be unable to properly mitigate mitochondrial stress. However, further investigation into the role of the UPRmt as a cardioprotective response to abnormal proteostasis needs to be investigated.

Several of the disorders associated with cardiomyopathy involve genes important for preprotein processing, such as *MIPEP*, *XPNPEP3*, *CLPB*, and *HTRA2*. However, some disorders involve genes important for preprotein processing but do not present with cardiomyopathy, such as mutations in mitochondrial-processing peptidase subunit alpha (*PMPCA*) [140] and mitochondrial-processing peptidase subunit beta (*PMPCB*) [141], which encode subunits of MPP, and inner mitochondrial membrane peptidase subunit 2 (*IMMP2L*) which encodes a subunit of the mitochondrial inner membrane peptidase (IMP) [142]. One possible reason for the lack of cardiac phenotypes observed in disorders of these genes could be that they do not involve MIP. MIP is the final peptidase for cleaving sequences of numerous proteins targeted to the matrix and IMM [143]. In fact, in *S. cerevisiae*, mutated MIP was found to impair oxidative metabolism and lead to a respiratory-deficient phenotype [144]. Thus, we can see how disorders impacting the ability to target or fold proteins that require MIP cleavage could drastically impact mitochondrial energy production and thereby cardiac function. From this, one could speculate that perhaps the reason why we see some disorders present with cardiomyopathy while others do not could lie in the fact that some processing peptidases play a greater role in the cleavage of signals targeted towards the ETC. Disruptions in proteostasis involving these peptidases would more drastically impact energy production, leading to oxidative stress and cardiomyopathy.

## 10. Conclusions and Future Directions

From this review, we see that cardiomyopathy can result from perturbed mitochondrial proteostasis through abnormalities in protein import, folding and maturation that disrupt OXPHOS, create imbalances in ROS and increase mitophagy and apoptosis. Genes important in these processes and that are associated with cardiomyopathy include *DNAJC19, MAGMAS, TIMM50, MIPEP, XPNPEP3, HTRA2, CLPB* and *HSPD1*. It is plausible that variants in other genes involved in proteostasis may cause less severe disruptions in OXPHOS and may not be significant enough to cause a cardiac phenotype. Alternatively, compensatory proteins or pathways may be activated in the heart to forestall the development of cardiomyopathy. These factors could also potentially explain the variation in severity of the disorders and why some patients present with cardiomyopathy, while others do not.

Considering the importance of the mitochondrial proteome in cell vitality, further investigation into the link between mitochondrial proteostasis and cardiomyopathy is needed. The diversity of the genes involved, and the diversity of patient phenotypes is not understood. While the genes discussed within this review point to important roles in the maintenance of mitochondrial proteostasis, all the disorders discussed are understudied. In addition to characterizing how mitochondrial proteostasis is disrupted, it will also be important to investigate the role of the UPRmt and other rescue pathways in potentially modifying the phenotype of these disorders. Furthering our understanding of how proteostasis is maintained and the factors leading to the activation of the UPRmt, particularly in mammals with cardiomyopathy, could lead to novel insights into mitochondrial biology.

## Figures and Tables

**Figure 1 ijms-23-03353-f001:**
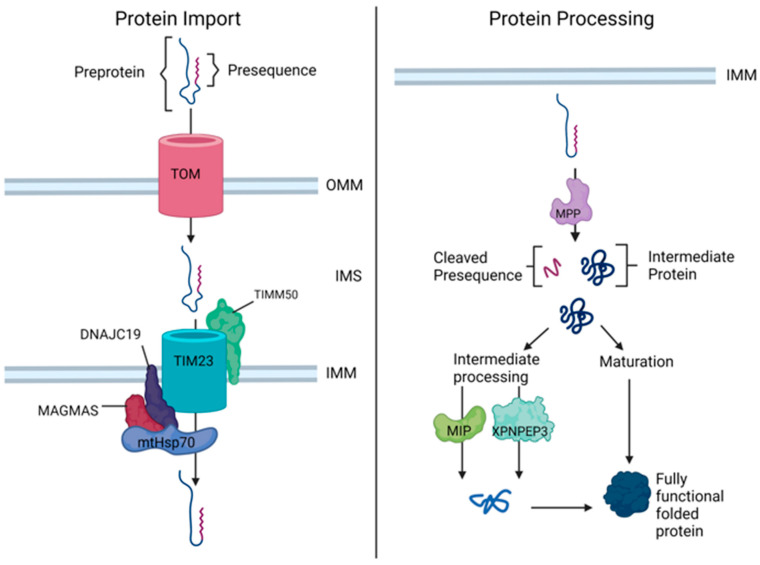
Proteins involved in mitochondrial protein homeostasis and associated with human cardiomyopathy. The translocase of the inner mitochondrial membrane (TIM23) is comprised of the translocase of the inner mitochondrial membrane 50 (TIMM50), mitochondrial import inner membrane translocase subunit TIM16 (MAGMAS), mitochondrial heat shock protein 70 (mtHsp70), and DnaJ heat shock protein family (Hsp40) member C19 (DNAJC19). Formation of the TIM23 complex pulls presequence proteins into the mitochondrial matrix. Mitochondrial processing peptidase (MPP), mitochondrial intermediate peptidase (MIP), and X-Pro aminopeptidase 3 (XNPEP3) perform processing and cleavage of presequence proteins once inside the matrix. Figure created in biorender.

**Figure 2 ijms-23-03353-f002:**
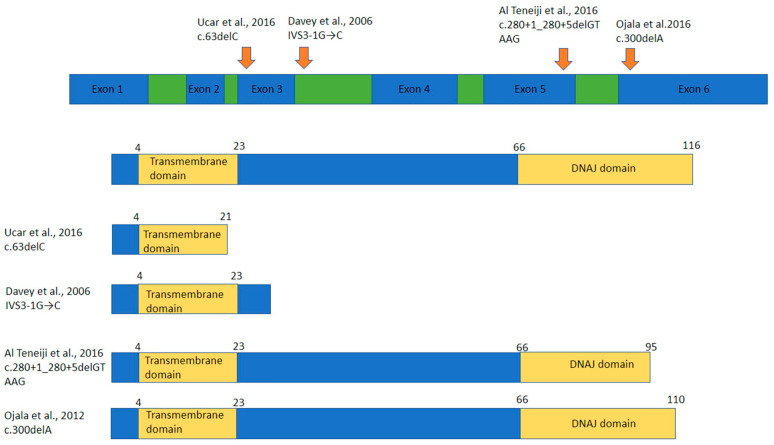
Described mutations in DnaJ heat shock protein family (Hsp40) member C19 (DNAJC19) [45,66,67,68]. Location of known mutations in DNAJC19 mRNA and their subsequent protein products. Green represents introns, blue represents exons, important domains in protein products are in yellow with residue numbers above each protein.

**Table 1 ijms-23-03353-t001:** Disorders presenting with cardiomyopathy that are involved in mitochondrial protein import into the inner mitochondrial membrane. Formal Online Mendelian Inheritance in Man (OMIM) report, protein function, and clinical characteristics are outlined.

Gene	Gene Name/Protein Function	Disorder	Cardiomyopathy	Clinical Characteristics
DNAJC19	DnaJ Heat Shock Protein Family Member C19/Modulates function and localization of mtHsp70	DCMA (Dilated Cardiomyopathy with Ataxia Syndrome): OMIM 610198	Dilated cardiomyopathy	Increased 3-methylglutaconic aciduria, growth failure, ataxia, gonadal dysgenesis, cardiac conduction defects
MAGMAS	Mitochondrial import inner membrane translocase subunit TIM16/Interacts with DNAJC19 to promote mtHsp70 activity	SMDMDM (Spondylometaphyseal dysplasia, Megarbane-Dagher-Melike type): OMIM 613320	Non-specific cardiomyopathy	Developmental delay, short stature, platyspondyly
TIMM50	Translocase of Inner Mitochondrial Membrane 50/TIM23 complex protein involved in recognition and sorting of incoming proteins	MGCA9 (3-Methylglutaconic Aciduria, Type IX): OMIM 617698	Cardiomyopathy	Seizures, hypotonia, delayed psychomotor development, increased 3-methylglutaconic aciduria

**Table 2 ijms-23-03353-t002:** Disorders presenting with cardiomyopathy that are involved in mitochondrial preprotein processing inside the mitochondrial matrix. Formal Online Mendelian Inheritance in Man (OMIM) report, protein function, and clinical characteristics are outlined.

Gene	Gene Name/Protein Function	Disorder	Cardiomyopathy	Clinical Characteristics
MIPEP	Mitochondrial intermediate peptidase/Final processing of nuclear-encoded proteins targeted to mitochondrial matrix or inner membrane	COXPD31 (Combined Oxidative Phosphorylation Deficiency 31): OMIM 617228	Left ventricular non-compaction cardiomyopathy, dilated cardiomyopathy, biventricular hypertrophic cardiomyopathy	Hypotonia, left ventricular non-compaction, developmental delay, cataracts
XPNPEP3	X-Prolyl Aminopeptidase 3/Peptidase that cleaves N-terminal amino acids	NPHPL1 (Nephronopthisis-like Nephropathy 1): OMIM 613159	HCM, DCM	Hypertension, tremor, renal insufficiency
HTRA2	HTRA2 (HtrA Serine Peptidase 2)/Involved in mediating apoptosis	MGCA8 (3-Methylglutaconic Aciduria, Type VIII): OMIM: 617248	Poor cardiac contractility	Seizures, cerebellar atrophy, increased 3-methylglutaconic aciduria

**Table 3 ijms-23-03353-t003:** Disorders presenting with cardiomyopathy resulting from mutations in genes encoding chaperones. Formal Online Mendelian Inheritance in Man (OMIM) report, protein function, and clinical characteristics are outlined.

Gene	Gene Name/Protein Function	Disorder	Cardiomyopathy	Clinical Characteristics
CLPB	Caseinolytic Mitochondrial Matrix Peptidase Chaperone Subunit B/Disaggregase	MEGCANN (3-methylglutaconic aciduria, type VII; MGCA7): OMIM 616271	DCM	Increased 3-methylglutaconic aciduria, neutropenia, cataracts, developmental delay, microcephaly, hypotonia
HSPD1	HSPD1 (Heat-Shock 60-kD Protein 1)/Involved in folding and degradation of misfolded proteins	Hypomyelinating Leukodystrophy 4 & Spastic paraplegia 13: OMIM 118190	No CM	Neurodegeneration, progressive spasticity, seizures, developmental arrest

## Data Availability

Not applicable.

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
