# Peer review of "Mitochondrial Protein Homeostasis and Cardiomyopathy"

_ijms, 2022, doi:10.3390/ijms23063353_

Round 1

Reviewer 1 Report

Wachoski-Dark and coauthors review links between loss of mitochondrial proteostasis and cardiomyopathies. This is an interesting topic, and the manuscript does an excellent job of integrating diverse evidence from model organisms through clinical studies of rare human diseases.  However, there are a few issues that need to be addressed, mostly related to clarity.

Major:

The first paragraph of the introduction is very dense and covers a lot of material. Please clarify and/or expand this paragraph.

Figure 1 needs to be reorganized and better labelled for clarity.  For example, the various peptides/folded proteins are ambiguous, and the arrows/processes are likely unclear to a non-expert reader.

The manuscript at multiple points links proteostasis to OXPHOS and ROS, often in the context of acute stress (ischemia-reperfusion is one example).  However, the relationship between acute stress and the development/progression of a chronic condition is vague and not thoroughly discussed.  Please discuss expand on this topic.

Minor:

Please remove the use of future tenses throughout.

There is a switch between active and passive voice that is distracting in a few places, for example the last paragraph of the introduction and the last paragraph of section 7.  Please consider switching to consistent active voice for clarity.

Please revise the first paragraph of section 2 for clarity.

Awkward or unclear sentences needing revision: lines 44, 74, 400, 409

Line 289: 25% of all substrates, or 25% of any substrate?

Line 313, does CLPB activate PARL?  I found this confusing.

Author Response

We thank the reviewer for their time and helpful suggestions. Reviewer comments are listed below with our responses in bold.

Major:

The first paragraph of the introduction is very dense and covers a lot of material. Please clarify and/or expand this paragraph.

We have revised and subdivided the Introduction to improve clarity.

Figure 1 needs to be reorganized and better labelled for clarity.  For example, the various peptides/folded proteins are ambiguous, and the arrows/processes are likely unclear to a non-expert reader.

The figure has been revised as requested (see attached).

The manuscript at multiple points links proteostasis to OXPHOS and ROS, often in the context of acute stress (ischemia-reperfusion is one example).  However, the relationship between acute stress and the development/progression of a chronic condition is vague and not thoroughly discussed. Please discuss expand on this topic.

We have now highlighted these different contexts (acute vs. chronic) throughout the manuscript where appropriate and discussed potential differences in the cellular response to acute vs. chronic stress (Section 9).

Minor:

Please remove the use of future tenses throughout.

There is a switch between active and passive voice that is distracting in a few places, for example the last paragraph of the introduction and the last paragraph of section 7. Please consider switching to consistent active voice for clarity.

Please revise the first paragraph of section 2 for clarity.

Awkward or unclear sentences needing revision: lines 44, 74, 400, 409

Line 289: 25% of all substrates, or 25% of any substrate?

Line 313, does CLPB activate PARL?  I found this confusing.

We have reviewed and corrected the manuscript for grammar and clarified the identified text.

Reviewer 2 Report

General comment:

This manuscript, entitled “Mitochondrial Protein Homeostasis and Cardiomyopathy,” authored by Wachoski-Dark et al., reviewed the mitochondrial diseases associated with cardiomyopathy in relation to mutations in genes that are important for mitochondrial proteostasis. A descriptive survey like this helps target these proteins for cardiac dysfunction-related diseases. This survey will provide essential links between cardiac stress and mutation in vital proteins during the development of heart failure. Still, it will also inspire therapeutic approaches for mitochondrial dysfunction. In my opinion, this is a valuable work and is suitable for publication in the International Journal of Molecular Sciences after the authors have addressed the following comments and questions:

Specific comments:

  • Discussing mitochondria without the role of ETC complex in relation to cardiac disease is incomplete – is there scope to add those?
  • What is the role of mitochondrial metalloproteins in cardiomyopathy?
  • Cite this - El-Hattab AW, Scaglia F. Mitochondrial Cardiomyopathies. Front Cardiovasc Med. 2016;3:25. Published 2016 Jul 25. doi:10.3389/fcvm.2016.00025?
  • How challenging diagnosing is mitochondrial disease?
  • What is evidence for mitochondrial disease states associated with cancer and diabetes?
  • The mitochondrial calcium (Ca2+) fluxes are the key regulator for cardiac function – what is the associate proteins mutation's role here?

Author Response

We thank the reviewer for their time and helpful suggestions. Reviewer comments are listed below with our responses in bold.

Specific comments:

Discussing mitochondria without the role of ETC complex in relation to cardiac disease is incomplete – is there scope to add those? What is the role of mitochondrial metalloproteins in cardiomyopathy?

We have added information about the ETC and mitochondrial metalloproteins to the manuscript.

Cite this - El-Hattab AW, Scaglia F. Mitochondrial Cardiomyopathies. Front Cardiovasc Med. 2016;3:25. Published 2016 Jul 25

This reference has been added.

How challenging diagnosing is mitochondrial disease?

Given the phenotypic heterogeneity and individual patient variability, the diagnosis of mitochondrial disease can be very challenging. We have added an additional paragraph (Section 5) to highlight this point.

What is evidence for mitochondrial disease states associated with cancer and diabetes?

These are extensive topics that could be the focus of separate reviews. We have chosen to restrict our manuscript to cardiomyopathy in order to highlight potentially useful disease-related connections. However, the have highlighted the connection between mitochondria, cancer and diabetes in the new Section 5.

The mitochondrial calcium (Ca2+) fluxes are the key regulator for cardiac function – what is the associate proteins mutation's role here?

We have added information about the role of calcium and cardiomyopathy.

Reviewer 3 Report

    The manuscript authored by Wachoski-Dark and colleagues is a review article dedicated to a description of a subset of known disorders of mitochondrial proteostasis that are associated with cardiomyopathy. Overall, the manuscript is well organized and clearly written and significantly contributes to the field of mitochondrial biology.

Minor:

Please address the following grammatical errors found in the manuscript:

Line 210: "These finding suggest" should be "these findings suggest".

Lines 297-298: "believed to implicated" should be "believed to be implicated".

Line 406: "ISR has shown be effective" should be "ISR has shown to be effective".

Author Response

We thank the reviewer for their time and helpful suggestions. Reviewer comments are listed below with our responses in bold.

Minor:

Please address the following grammatical errors found in the manuscript:

Line 210: "These finding suggest" should be "these findingsuggest".

Lines 297-298: "believed to implicated" should be "believed to be implicated".

Line 406: "ISR has shown be effective" should be "ISR has shown to be effective".

We have reviewed and corrected the manuscript for grammar.
